# GPR Image Clutter Suppression Using Gaussian Curvature Decomposition in the PCA Domain

**Qibin Su** **, Beizhen Bi, Pengyu Zhang, Liang Shen, Xiaotao Huang and Qin Xin \***

The College of Electronic Science, National University of Defense Technology, Changsha 410073, China
* Correspondence: xinqin@nudt.edu.cn

**Abstract:** Ground penetrating radar (GPR) is one of the most generally used underground sensing equipment, but it is frequently contaminated by clutter and noise during data acquisition, which has a significant impact on the detection performance of buried targets. The purpose of this letter is to present a novel clutter suppression method based on the principal component Gaussian curvature decomposition (PCGCD). First, the GPR B-scan data are divided into different sub-components using principal component analysis (PCA). Then, a Gaussian curvature decomposition (GCD) method is proposed, which can be applied to PCA domain subspaces to recover more target structure information from random noise. The PCGCD method's performance is evaluated using both numerical simulation and real-world GPR datasets. The visualization and quantitative results demonstrated our method's superiority in protecting the underground target structure, removing complex random noise, and improving the detection ability of buried targets.

**Keywords:** ground penetrating radar (GPR); clutter suppression; Gaussian curvature filter (GCF); principal component analysis (PCA)

## 1. Introduction

Ground penetrating radar (GPR) is a non-destructive and effective technology for detecting underground media using electromagnetic waves. However, in practice, the visibility of underground targets in GPR B-scan images is usually blocked due to the interference of various types of clutter. GPR clutter primarily consists of the coupling effect between the transmitting and receiving antennas, ground surface reflection, dispersion of the underground medium, and instrument noise [1,2], all of which obscure the target response and severely damage the target detection capability, particularly when the target response overlaps with random noise and clutter. Thus, clutter suppression is not only a key technology in GPR data preprocessing but also a critical link in improving target detection and feature extraction performance.

The simplest clutter suppression technique is mean subtraction (MS). MS can reduce homogeneous ground bounce to a large extent, but it is ineffective against clutter in soil with high moisture content and random noise [3]. Subspace-based clutter removal methods include singular value decomposition (SVD) [4], principal component analysis (PCA) [5] which decompose the B-scan data into clutter and target subspaces, and those methods rely heavily on subspace differentiation. Aside from traditional approaches, such as mean value subtraction or subspace-based approaches, many low ranks and sparse decomposition methods, such as non-negative matrix factorization (NMF) [6], robust non-negative matrix factorization (RNMF) [7], and robust PCA (RPCA) [8] express GPR B-scan data as the sum of a low-rank and a sparse matrix. The low-rank matrix and the sparse matrix, respectively, contain clutter and target responses [9]. However, both RPCA and RNMF necessitate parameter pre-setting, which is time-consuming. In the method based on morphological component analysis (MCA) [10], the curvelet and undecimated discrete wavelet transform dictionaries are used to describe clutter and target components but the performance of

MCA is largely dependent on artificially constructed adaptive dictionary. Deep-learning-based methods [11–14] have been employed in solving GPR clutter removal problems; however, they are only trained on the simulated or synthetic datasets, generally degrading the clutter removal capability in real-world GPR radargrams, because the real-world clutter distributions is complex and diverse [12].

Gong in [15] recently presented the Gaussian curvature filter (GCF), a powerful spatial filtering method with edge preservation and image denoising functions. The GCF has been used to solve various image processing problems, including image enhancement, fusion, feature extraction, and automatic visual detection [16–20]. In this letter, we proposed a new GPR clutter suppression method called principal component Gaussian curvature decomposition (PCGCD) to suppress clutter and random noise while obtaining underground target characteristics. To begin, the PCA is used to divide the GPR B-scan data into different components before removing the first principal component. The GCD then decomposes the remaining GPR image into multiple detailed layers to extract the detailed texture features. Finally, the PCA is applied to each image layer to suppress residual clutter and obtain the processed GPR image. We compared our method to other well-known methods in the literature to demonstrate its superiority. The peak signal-to-noise ratio (PSNR) and structural similarity (SSIM) in simulation datasets, as well as visual results in real datasets, are used to demonstrate the performance analysis. The proposed method is tested on both simulated and real-world hybrid GPR data [14], and the experimental results show that the proposed method is superior.

The rest of this article is divided into sections. The methodology is detailed in Section 2. Section 3 assesses the PCGCD's efficiency by analyzing experimental results from simulation and real-world GPR image datasets. Section 4 finally presents the conclusions drawn from the results.

## 2. Methodology

### 2.1. Principal Component Analysis Based Clutter Suppression

Aside from clutter, various types of random noise interference are present in the GPR detection process [13]. Useful information is easily obscured by clutter and random noise. To better interpret such images, clutter suppression, and image filtering are required. An $m \times n$ rectangular matrix $\mathbf{X}$ is used to represent a GPR B-scan image, where $m$ and $n$ represent the depth and number of antenna locations, respectively (down-track). It is the sum of the target, clutter, and noise subspaces:

$$\mathbf{X} = \mathbf{X}_{target} + \mathbf{X}_{clutter} + \mathbf{X}_{noise} \tag{1}$$

PCA is a multivariate statistical analysis method that processes multidimensional data efficiently. It transforms a set of possibly correlated variables into linearly uncorrelated variables using orthogonal transformation [5,21]. The $N$ principal components of the GPR B-scan image $\mathbf{X}$ can be shown as follows for clutter suppression using PCA:

$$\overline{\mathbf{Y}} = \mathbf{A}^{\mathrm{T}}\overline{\mathbf{X}} \tag{2}$$

where $\overline{\mathbf{X}}$ denotes the centralized matrix of $\mathbf{X}$ and $\overline{\mathbf{Y}}$ denotes the vector principal components of $\overline{\mathbf{X}}$. The covariance matrix of $\overline{\mathbf{X}}$ are provided in descending order in the transformation matrix bf $\mathbf{A}$ , whose dimensions is $m \times n$. $\mathbf{A} = [\mathbf{A}_1, \mathbf{A}_2, ..., \mathbf{A}_m]$ where $\mathbf{A}_1$ is the eigenvector corresponding to the largest eigenvalue.

PCA's ability to de-correlate fully is an important property. Because the response of clutter is more continuous than the response of the target or soil, the first principal component is usually identified as clutter, whereas the noise energy is evenly distributed across the entire GPR dataset. As a result, PCA may frequently divide a GPR image into three parts as follows:

$$\mathbf{X} = \mathbf{X}_{clutter} + \mathbf{X}_{target+noise} = \mathbf{A}_1 \mathbf{A}_1^T \overline{\mathbf{X}} + \sum_{i=2}^{N} \mathbf{A}_i \mathbf{A}_i^T \overline{\mathbf{X}} \tag{3}$$

### 2.2. The Gaussian Curvature Filter Based Denoising

For each variational formulation, a minimizing function to an energy functional must be realized.

$$E(U) = E_{\phi_0}(U, I) + \lambda E_{\phi_1}(U) \tag{4}$$

The total energy $E(U)$ is made up of two parts: the data-fitting energy $E_{\phi_0}(U, I)$, which measures the degree of fit of $U$ to the image $I$, and the regularization energy $E_{\phi_1}(U) \geqslant 0$, which formalizes prior knowledge of $U$. $\lambda$ is the scalar regularization coefficient for weighting these two parts.

The progression of the image energy functional evolution is shown in Figure 1. The data-fitting energy $E_{\phi_0}$ is increasing, while the total energy $E$ and regularization energy $E_{\phi_1}$ are decreasing. As a novel algorithm, the GCF implicitly employs differential geometry theory to optimize regularization energy. It is assumed that the surface of an ideal noise-free image is developable and that the developed surface's Gaussian curvature is zero everywhere [16]. Making the image "as developable as possible" reduces GC and avoids explicit GC calculation. The Gaussian curvature filtering procedure is divided into five steps, as demonstrated in Figure 2:

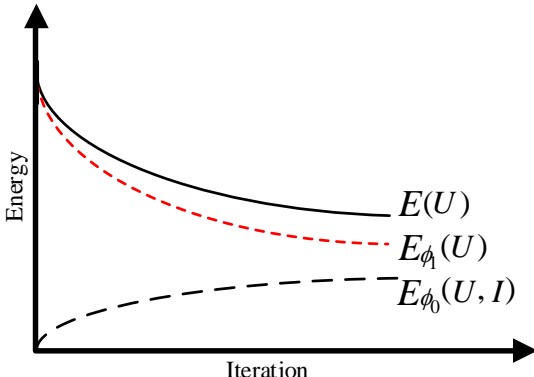

**Figure 1.** Regularization-dominated variational models.

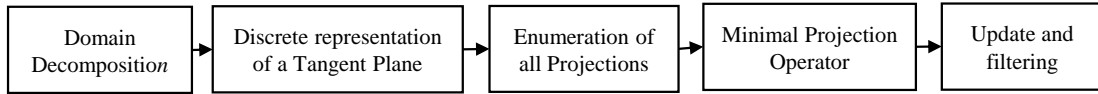

**Figure 2.** Flow chart of the Gaussian curvature filter.

- *Domain Decomposition*: The original image is decomposed into four subsets: white circles $W_C$, white triangles $W_T$, black circles $B_C$, and black triangles $B_T$. This decomposition ensures that neighbors are in separate subsets and eliminates pixel dependency, as demonstrated in Figure 3a;
- *Discrete representation of Tangent Plane*: Triangle windows are chosen for their ease of projection to construct a hypothetical projection tangent plane, as shown in Figure 3b. After obtaining this tangent plane, we compute the distance $d_k$ as shown in Figure 3c, and then project the center pixel onto the triangle plane's edge;
- *Enumeration of all Projections*: It is necessary to locate the tangent plane with the smallest $|d_k|$. We count the number of tangent triangles that can exist in a $3 \times 3$ window. There are 12 such triangles and 8 different $|d_k|$ [16], some of which are shown in Figure 3d–f;
- *Minimal Projection Operator*: Computing $\{d_k, k = 1, 2, \ldots, 8\}$, we employ the smallest absolute distance $|d_m| = min\{d_k, k = 1, 2, \ldots, 8\}$ as the minimum projection of the current intensity. Then, we let $\hat{U}(p, q) = U(p, q) + d_m$. This operator is summarized in Algorithm 1;

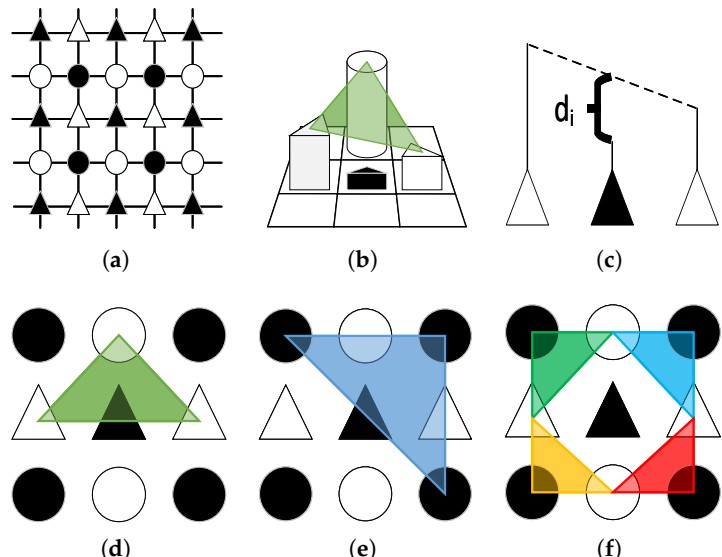

**Figure 3.** Structure of the Gaussian curvature optimization. (**a**) Illustration of the minimum vertex coloring domain decomposition; (**b**) A tangent plane; (**c**) Illustration of the projectionb distance; (**d**–**f**) The twelve possible tangent planes in a 3 × 3 neighborhood.

- *Update and filtering*: All of the pixels $W_C$, $W_T$, $B_C$, and $B_T$ of the minimal projection operator are iterated. The Gaussian curvature filter can be generated as:

$$I = G_c(U(x)), x \in \{W_T, W_C, B_T, B_C\} \tag{5}$$

The GCF effectively combines the discreteness of image data and the continuity of differential geometry by researching the surface rather than calculating the curvature. In comparison to other filtering algorithms, the GCF has no parameters. Furthermore, the requirement for second-order variability on the image surface has been removed, allowing for sharp corners and edges in the image while ideally protecting its edges [20]. Therefore, the GCF can remove random noise while protecting the integrity of underground target features.

---

**Algorithm 1:** Minimal projection operator

---

**Input:** $U(p, q)$
**Output:** $\hat{U}(p, q) = U(p, q) + d_m$
1: $d_1 = (U(p-1, q) + U(p+1, q))/2 - U(p, q)$
2: $d_2 = (U(p, q-1) + U(p, q+1))/2 - U(p, q)$
3: $d_3 = (U(p-1, q-1) + U(p+1, q+1))/2 - U(p, q)$
4: $d_4 = (U(p-1, q+1) + U(p+1, q-1))/2 - U(p, q)$
5: $d_5 = U(p-1, q) + U(p, q-1) - U(p-1, q-1) - U(p, q)$
6: $d_6 = U(p-1, q) + U(p, q+1) - U(p-1, q+1) - U(p, q)$
7: $d_7 = U(p, q-1) + U(p+1, q) - U(p+1, q-1) - U(p, q)$
8: $d_8 = U(p, q+1) + U(p+1, q) - U(p+1, q+1) - U(p, q)$
9: find $d_m$, such that $|d_m| = min\{|d_k|, k = 1, 2, \ldots, 8\}$

---

## 3. Proposed Method

### 3.1. Gaussian Curvature Decomposition

In this letter, we introduced the GCF to the idea of bidimensional empirical mode decomposition [22] and proposed the GCD. Its principle is to adaptively decompose a given original image into multiple detail layers, also known as intrinsic mode functions

(IMF). This is known as sifting. The sifting process of a GPR image $I(p,q)$ is as follows (Algorithm 2):

---

**Algorithm 2:** Sifting process of Gaussian Curvature Decomposition

---

    **Input:** Original image $I(p,q)$
    **Output:** $I(p,q) = \sum_{i=1}^{N} IMF_i(x,y) + Ir(p,q)$, where N is the number of *IMFs*
    **Initialization:** $Ir_{i,j-1}(p,q) = I(p,q), i = 1, j = 1, \varepsilon = 1, N = 3$.
    **Main iteration:**
    **1:** Extracting extrema of $Ir_{i,j-1}$, according to the Algorithm 1 upper envelope
    $Imax_{i,j-1} = Ir_{i,j-1} + d_M, |d_M| = \max\{|d_k|, k = 1,2,\ldots,8\}$ and lower envelope
    $Imin_{i,j-1} = Ir_{i,j-1} + d_m, |d_m| = \min\{|d_k|, k = 1,2,\ldots,8\}$ to compute the mean
    envelope as: $I_{i,j}(p,q) = (Imax_{i,j-1}(p,q) + Imin_{i,j-1}(p,q))/2$
    **2:** Calculate $H_{i,j-1}(p,q)$ by $H_{i,j-1}(p,q) = Ir_{i,j-1}(p,q) - M_{i,j-1}(p,q)$.
    **3:** We utilize the standard deviation of $H_{i,j-1}$ to determine the stopping criterion.
    If $std2(H_{i,j-1}) \le \varepsilon$, new $IMF_i(p,q) = H_{i,j-1}(p,q)$ is obtained, after which the
    next step is executed, otherwise update $j = j + 1, Ir_{i,j-1}(p,q) = H_{i,j-2}(p,q)$ , and
    repeat step 1–3.
    **4:** Obtain the new residual:$i = i + 1, Ir_{i,0}(p,q) = Ir_{i-1,0}(p,q) - IMF_{i-1}(p,q)$, if
    $i < N$, repeat step 1–4, otherwise the GCD procedure is terminated, the residual
    $Ir(p,q) = Ir_{i,0}(p,q)$ is obtained.

---

The GCF computes the approximate continuous smooth surface of the raw GPR image, which can be used as the envelope surface of the empirical mode decomposition to decompose the two-dimensional natural mode function image. This process can continuously remove the influence of the mean value and highlight the operation of local extreme values, making the image's features more prominent. At the same time, the texture and edges are enhanced.

### 3.2. PCGCD Based Clutter and Noise Suppression Method

The number of different subspaces in PCA technology is difficult to determine in strong random noise and multi-target environments. Therefore, it is far from possible to suppress clutter in GPR images using a single PCA technology. As a result of combining PCA with the proposed GCD, we developed a new GPR clutter suppression method that combines the PCA domain and spatial filtering. The flowchart in Figure 4 shows the proposed method for removing clutter and noise.

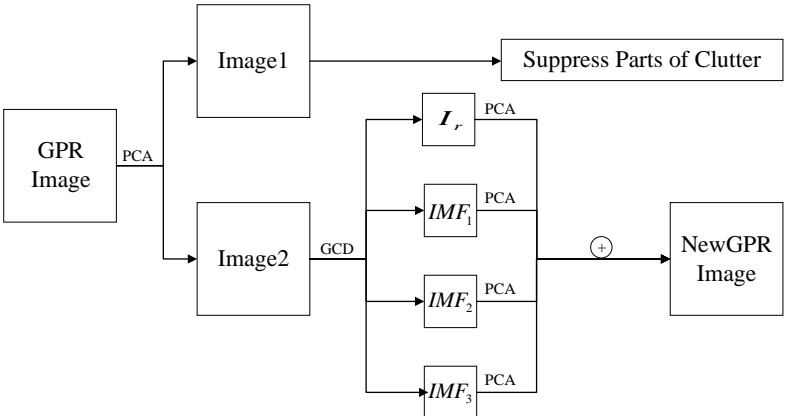

**Figure 4.** Flow chart of PCGCD-based clutter suppression method.

## 4. Experiments

This section validates the efficiency of the clutter and noise suppression methods using both simulation and actual GPR datasets. The proposed methods are compared to the

MS, SVD, NMF, PCA, RNMF, and RPCA. The quantitative and visual findings are shown for comparison.

### 4.1. Simulation Datasets Results

We generated simulation datasets using the gprMax simulation software [23]. Because gprMax can also generate reference images, the simulated data allow us to conduct a quantitative study in addition to a visual one. As a result, the quantitative results rely on the peak signal-to-noise ratio (PSNR) and SSIM indices.

The SSIM is a useful indicator for comparing the similarity of structure information before and after clutter suppression. Generally calculated as:

$$SSIM = \frac{(2\mu_x\mu_y + c_1)(2\sigma_{xy} + c_2)}{\left(\mu_x^2 + \mu_y^2 + c_1\right)\left(\sigma_x^2 + \sigma_y^2 + c_2\right)} \tag{6}$$

where $\mu_x$, $\mu_y$, $\sigma_x$ and $\sigma_y$ denote the mean and variance of the image before and after suppression, $\sigma_{xy}$ denotes the covariance of the two images, and $c_1$ and $c_2$ denote constants used to ensure stability. The normal range of SSIM values is 0–1. The closer the SSIM value is to 1, the more similar the structure of the two images is.

PSNR is given as

$$PSNR(dB) = 10 \times log\frac{1}{MSE} \tag{7}$$

and the mean square error (MSE) can be expressed as

$$MSE = \frac{1}{m \times n} \sum_{p=1}^{m} \sum_{q=1}^{n} (X(p,q) - Y(p,q))^2 \tag{8}$$

The experimental design of the simulation dataset is as follows. A commercial antenna Ricker from the gprMax library with a frequency of 2-GHz. Different soil types and target materials are used in the simulations to simulate various scenarios. We simulated two types of roads: cement and asphalt. As shown in Figure 5, four pipe targets with a radii of 3 cm and thicknesses of 3 mm are placed at the 10 cm depth of the road surface. The size of the simulated image is 849 × 480. Table 1 lists the electromagnetic properties of the materials.

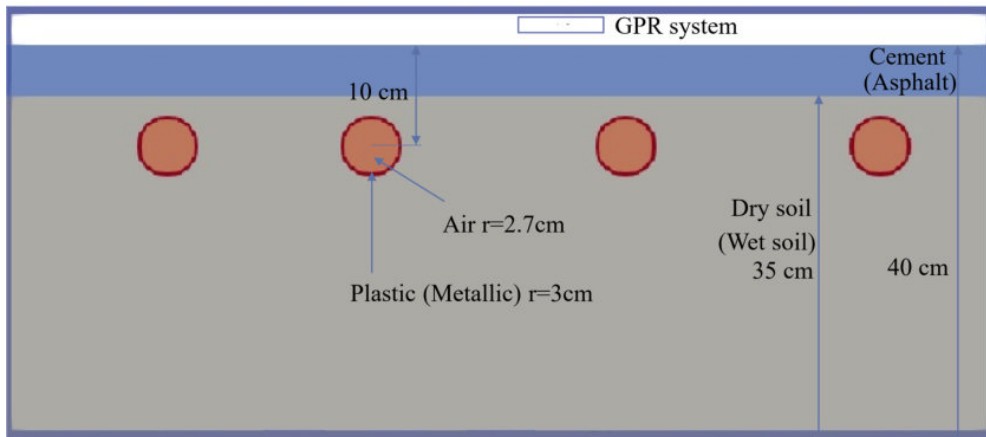

**Figure 5.** Simulation model.

We added random noise with a noise ratio of 15 dB to the simulation model, which placed four metal pipe targets in a cement scene to show the real scene (Cement + Dry soil). Figure 6a,b demonstrate the simulation's original and noisy images. Figure 6c shows the reference image. Figure 6d–j show the extracted target image by the MS, SVD, PCA, RPCA, NMF, RNMF, and PCGCD, respectively.

**Table 1.** Electromagnetic properties of the material.

| Meterial | Relative Permittivity (F/m) | Conductivity (S/m) |
|---|---|---|
| Cement | 7 | 0.001 |
| Asphalt | 5 | 0.001 |
| Dry soil | 10 | 0.01 |
| Wet soil | 12 | 0.01 |
| Plastic | 3 | 0.01 |
| Metallic | 3.1 | $2.3 \times 10^7$ |

The SVD is implemented for the simulated dataset results by removing the largest singular value of the B-scan matrix, and there is no parameter selection for the PCA and NMF. The critical $\lambda$ parameters are set to $5 \times 10^{-2}$ for RNMF and $1 \times 10^{-7}$ for RPCA, with iterations set to 1000 in both cases. For RNMF, the rank $r$ is set to $k = 1$ because the signal strength of the clutter outweighs the signal reflected from the target.

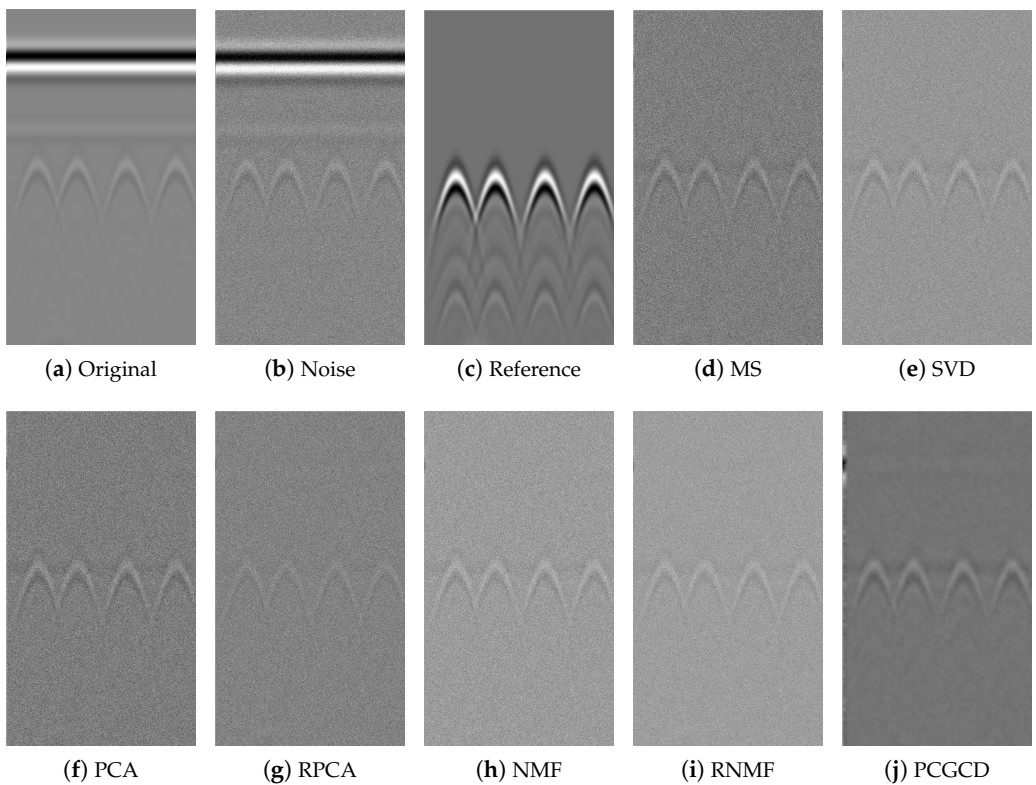

| (**a**) Original | (**b**) Noise | (**c**) Reference | (**d**) MS | (**e**) SVD |
|---|---|---|---|---|

| (**f**) PCA | (**g**) RPCA | (**h**) NMF | (**i**) RNMF | (**j**) PCGCD |
|---|---|---|---|---|

**Figure 6.** Clutter suppression results for the simulated data with different methods.

As shown in Figure 6, the ground reflection is removed in all methods. The influence of random noise is difficult to eliminate using traditional MS and PCA. The SVD, RPCA, and NMF results are nearly identical for this simple scenario. RNMF outperforms PCGCD in comparison methods, while PCGCD provides a clearer background in the visual sense, removing most of the random noise and protecting the target's edge. The outline of the targets can be seen clearly in Figure 6j.

PSNR and SSIM were calculated using the reference image obtained by gprMax, and the targets of two materials, plastic and metallic, were buried in different simulation scenarios. The quantitative results are shown in Tables 2 and 3. The evaluation times are shown in Table 4 and all methods are run on an AMD Ryzen 74800H with Radeon Graphics 2.90 GHz, 16 GB, and Nvidia GTX1660Ti in a Windows 11 64-bit environment.

**Table 2.** **PSNR** (dB) results for the plastic and metallic piplines buried in different scenarios.

| | Plastic Pipeline | | | | | | | Metallic Pipeline | | | | | | |
|---|---|---|---|---|---|---|---|---|---|---|---|---|---|---|
| Scenario | MS | SVD | NMF | PCA | RNMF | RPCA | **PCGCD** [1] | MS | SVD | NMF | PCA | RNMF | RPCA | **PCGCD** |
| Cement+Dry soil | 18.39 | 17.63 | 17.29 | 18.42 | 17.23 | 19.07 | **19.63** | 20.20 | 14.67 | 14.51 | 20.13 | 14.04 | 17.40 | **23.32** |
| Cement+Wet soil | 17.82 | 17.17 | 16.82 | 17.80 | 16.76 | 18.57 | **19.27** | 19.78 | 14.67 | 14.71 | 19.64 | 14.31 | 17.65 | **22.78** |
| Asphalt+Dry soil | 17.75 | 17.79 | 17.54 | 17.74 | 17.50 | 19.04 | **19.37** | 19.91 | 14.17 | 13.83 | 19.83 | 13.385 | 17.40 | **22.45** |
| Asphalt+Wet soil | 17.86 | 16.78 | 16.79 | 17.74 | 16.74 | 18.77 | **19.34** | 19.53 | 14.25 | 14.35 | 19.53 | 13.93 | 17.08 | **22.25** |

[1] Mark the results of our method in bold.

**Table 3.** SSIM results for the plastic and metallic pipelines buried in different scenarios.

| | Plastic Pipeline | | | | | | | Metallic Pipeline | | | | | | |
|---|---|---|---|---|---|---|---|---|---|---|---|---|---|---|
| Scenario | MS | SVD | NMF | PCA | RNMF | RPCA | **PCGCD** | MS | SVD | NMF | PCA | RNMF | RPCA | **PCGCD** |
| Cement+Dry soil | 0.138 | 0.193 | 0.157 | 0.137 | 0.274 | 0.147 | **0.786** | 0.164 | 0.221 | 0.178 | 0.161 | 0.323 | 0.144 | **0.831** |
| Cement+Wet soil | 0.130 | 0.182 | 0.149 | 0.128 | 0.263 | 0.145 | **0.762** | 0.148 | 0.203 | 0.162 | 0.144 | 0.296 | 0.149 | **0.819** |
| Asphalt+Dry soil | 0.130 | 0.194 | 0.156 | 0.128 | 0.274 | 0.141 | **0.777** | 0.169 | 0.229 | 0.181 | 0.169 | 0.328 | 0.146 | **0.829** |
| Asphalt+Wet soil | 0.125 | 0.179 | 0.146 | 0.124 | 0.256 | 0.144 | **0.761** | 0.156 | 0.210 | 0.174 | 0.152 | 0.317 | 0.150 | **0.817** |

**Table 4.** Running time evaluation of different clutter removal methods in simulate GPR image.

| **Method** | MS | SVD | NMF | PCA | RNMF | RPCA | PCGCD |
|---|---|---|---|---|---|---|---|
| **Time (s)** | 0.0015 | 0.0024 | 0.0500 | 0.0160 | 8.1356 | 3.1347 | 0.8615 |

In the simulation models, PCGCD produces the highest PSNR results. RPCA and PCA come in second place in the plastic and metal categories, respectively. In both cases, SVD and RNMF produce the lowest scores. The highest PSNR and SSIM indexes fully reflect PCGCD's clutter suppression and edge protection performance. Furthermore, RNMF and RPCA require more processing time of any methods. The PCGCD based method takes longer than the other classical methods, but the result is superior.

As a comparison experiment, we added 10 dB random noise to the simulate model. As a result, the average PSNR and SSIM of plastic and metal pipes in various scenarios are shown in Tables 5 and 6, respectively.

Again, in different noise intensities, the PCGCD-based method achieves higher PSNR and SSIM indexes than other methods. Because the target response in GPR B-scan data is sparse, the PSNR of the image processed by the same method will not be significantly improved for different noise intensities. Although SSIM is sensitive to noise changes, targets are easily buried in low SNR situations.

**Table 5.** The average of **PSNR** (dB) results with different SNR.

| | Plastic Pipeline | | | | | | | Metallic Pipeline | | | | | | |
|---|---|---|---|---|---|---|---|---|---|---|---|---|---|---|
| SNR (dB) | MS | SVD | NMF | PCA | RNMF | RPCA | **PCGCD** | MS | SVD | NMF | PCA | RNMF | RPCA | **PCGCD** |
| 10 | 17.38 | 17.47 | 17.42 | 17.38 | 17.90 | 18.81 | **20.77** | 17.07 | 17.08 | 16.86 | 17.08 | 17.31 | 18.29 | **20.80** |
| 15 | 17.96 | 17.79 | 17.12 | 17.92 | 17.05 | 18.86 | **21.53** | 19.86 | 14.44 | 14.35 | 19.78 | 13.91 | 17.38 | **22.63** |

**Table 6.** The average of (**SSIM**) results with different SNR.

| | Plastic Pipeline | | | | | | | Metallic Pipeline | | | | | | |
| --- | --- | --- | --- | --- | --- | --- | --- | --- | --- | --- | --- | --- | --- | --- |
| SNR (dB) | MS | SVD | NMF | PCA | RNMF | RPCA | **PCGCD** | MS | SVD | NMF | PCA | RNMF | RPCA | **PCGCD** |
| 10 | 0.077 | 0.080 | 0.076 | 0.070 | 0.084 | 0.138 | **0.510** | 0.077 | 0.080 | 0.078 | 0.079 | 0.099 | 0.167 | **0.583** |
| 15 | 0.131 | 0.187 | 0.129 | 0.152 | 0.267 | 0.144 | **0.772** | 0.159 | 0.216 | 0.174 | 0.156 | 0.316 | 0.147 | **0.824** |

### 4.2. Real Dataset Results

We collected real GPR data on asphalt pavements in this section. The experimental equipment includes a 3D ground penetrating radar, a vehicle carrier, a step frequency radar transmitter, a scanning step size of 0.07 m, and a scanning depth of 2 m. The critical $\lambda$ parameters are set to $3 \times 10^{-2}$ for RNMF and $1 \times 10^{-7}$ for RPCA, with iterations set to 1000 in both cases.

Figure 7b shows the asphalt road's raw B-scan image. This road's underground media is layered and has many hyperbolic features. The red box represents the high clutter interference caused by ground reflection, and the yellow section represents the vertical stripe radio frequency interference. Part of the ground reflection was suppressed by the MS, SVD, and PCA. NMF and RNMF images are not clean, and clutter is not effectively suppressed. The performance of RPCA and PCGCD is comparable, but we can see that the texture features are more prominent in Figure 7i.

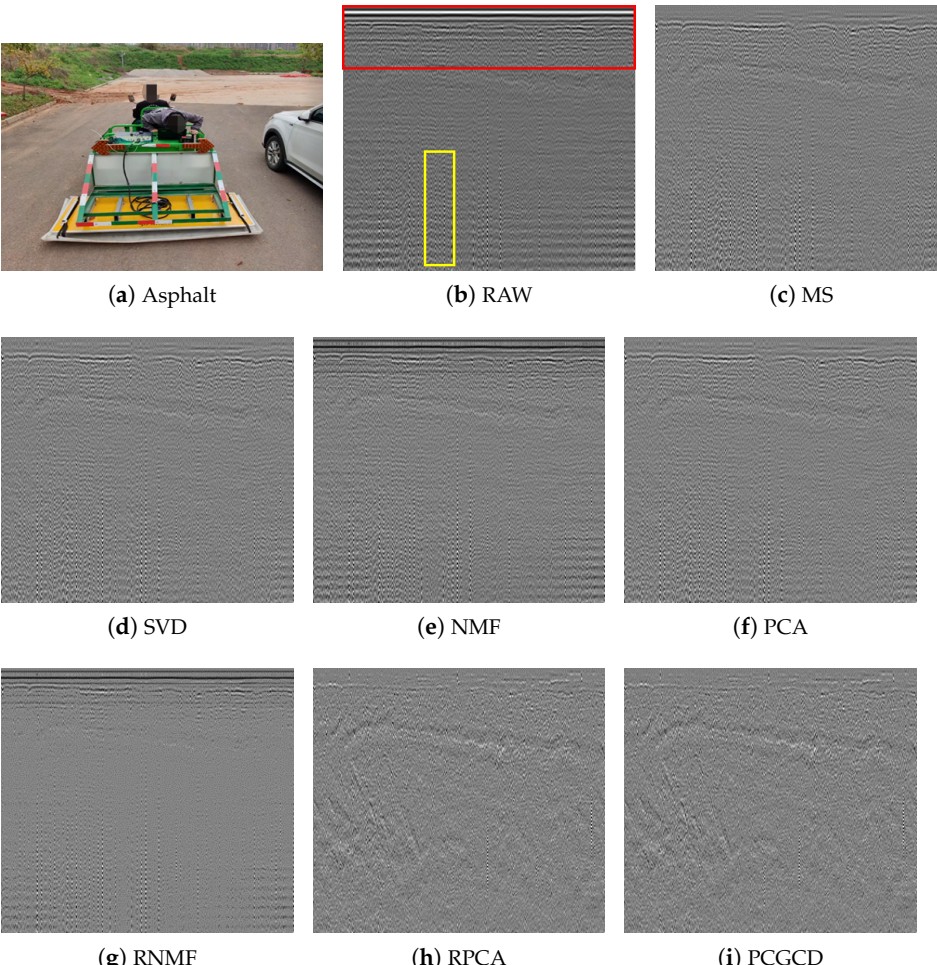

**Figure 7.** Asphalt road and reasults.

We can only analyze the data we collected through visual effects because it lacks reference datasets. To validate the experimental results, we used real-world GPR hybrid data in [14]. In a controlled environment, clutter-only radargrams were employed to collect a hybrid raw GPR image, which is then combined with a simulated clutter-free image by gprMax. The simulated clutter-free image also can be used as a reference image. The processed image results are shown in Figure 8, and the PSNR, SSIM, and evaluation times are shown in Table 7.

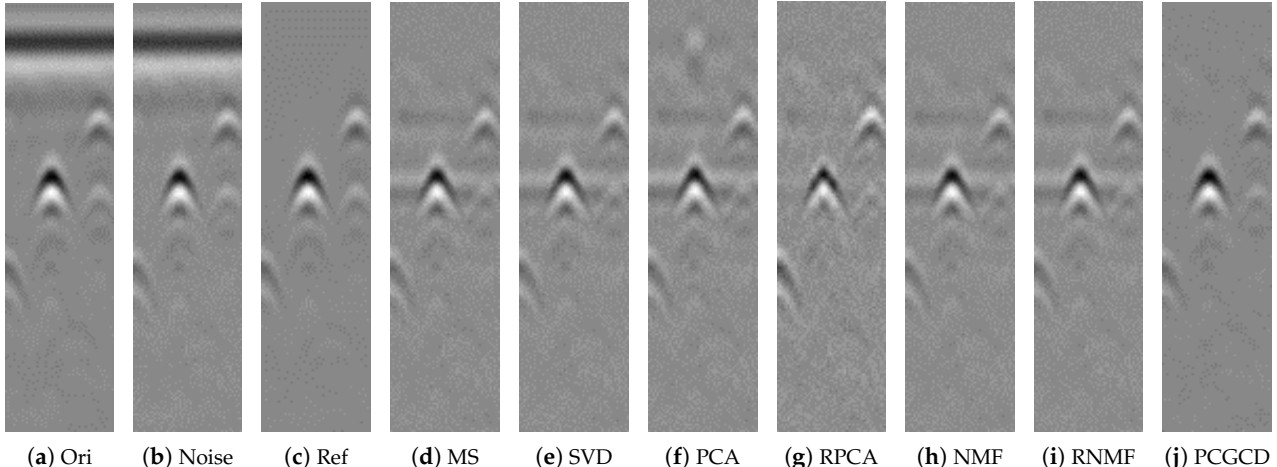

**(a)** Ori   **(b)** Noise   **(c)** Ref   **(d)** MS   **(e)** SVD   **(f)** PCA   **(g)** RPCA   **(h)** NMF   **(i)** RNMF   **(j)** PCGCD

**Figure 8.** Clutter suppression results for a hybrid raw GPR image with different methods.

**Table 7.** Running time evaluation of different clutter removal methods in simulate GPR image.

| Method | MS | SVD | NMF | PCA | RNMF | RPCA | PCGCD |
|---|---|---|---|---|---|---|---|
| PSNR | 30.88 | 31.01 | 31.01 | 29.62 | **35.36** [1] | 26.90 | 29.99 |
| SSIM | 0.722 | 0.722 | 0.710 | 0.681 | 0.830 | 0.513 | **0.920** [2] |
| Time (s) | 0.0051 | 0.0023 | 0.0456 | 0.0075 | **0.4181** [3] | 0.1348 | 0.0350 |

[1] RNMF has the highest PSNR. [2] PCGCD has the highest SSIM. [3] RNMF takes longer.

As shown in Figure 8, residual clutter is observed in the results of MS, SVD, PCA, and RPCA. Except for the PCGCD result, which is very similar to the reference image Figure 8c, all the resultant images have obvious random noise. Table 7 shows that the PCGCD method still has advantages in SSIM. The RNMF outperformed the PCGCD in PSNR, but its computing time is an order of magnitude longer.

## 5. Conclusions

A new GPR clutter suppression method, PCGCD, is proposed in this letter. This method combines the benefits of spatial pixel filtering and subspace technology, which can suppress clutter, eliminate interference, remove noise, and preserve edges. The PCGCD improves the PSNR and SSIM of GPR images while also highlighting the target contour, which is extremely useful for feature point detection and extraction in GPR images. The visual and quantitative results of numerical simulation and real GPR datasets demonstrate the proposed method's superiority over commonly used clutter suppression methods.

**Author Contributions:** Q.S.'s contribution is conceptualization, investigation, methodology, and writing original manuscript. B.B. provided valuable suggestions for the research. P.Z. and L.S. coordinated reading, analyzing and categorizing the articles reviewed in this study. X.H. and Q.X.'s coordinated conceptualization, methodology and resources. All authors have read and agreed to the published version of the manuscript

**Funding:** This research received no external funding.

**Data Availability Statement:** Not applicable.

**Conflicts of Interest:** The authors declare no conflict of interest.

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
