# Peer review of "GPR Image Clutter Suppression Using Gaussian Curvature Decomposition in the PCA Domain"

_remotesensing, doi:10.3390/rs14194879_

Round 1

Reviewer 1 Report

This paper proposed a PCGCD method to suppress GPR radargram clutter and noise.

(1)    I think the contrast of current denoising methods such as RPCARNMF and deep learning-based methods, including denoising effects and time consuming needs to be added;

(2)   In the Introduction, some new denoising methods, including deep learning-based framework needs to be added;

(3)   In page 7, “we added random noise with noise ratio of 20dB to the simulate model”. More SNR experiments are needs , for example, the denoising results of 10, 5, -5, -10dB SNR datasets.

(4)   In page 7, Line 162, “In Fig. ??,” please check.

Reviewer 2 Report

The manuscript fails at highlighting the significance and impact of the proposed methodology. Clutter reduction in GPR signal processing is a widely discussed topic, and several methodologies (i.e. MS, SVD, NMF and PCA) are already in place to tackle such problem, as discussed by the Authors themselves. The proposed methodology, together with the case studies, fail to demonstrate their significant contribution to the knowledge. The benefits obtained with the proposed method compared to traditional methods are in fact minimal, based on the proposed quantitative analysis. On a separate note, the manuscript style is not convincing, as the sentences are very short and unrelated. In addition, the Introduction reports a limited number of citations, and consequently several statements are not supported by references. Considering the above, I do not recommend the publication of the manuscript on Remote Sensing.

Reviewer 3 Report

This paper proposes a new method for mitigate/supress clutter in GPR. The topic is of interest for the GPR community due to the strong influence of clutter in the detection capabilities. However, the paper must be improve to deserve publication.

My main concern with this article is that it is not clear that the proposed approach outperforms other methods in the literature. First of all, the authors do not explain the parameters involved when applying other methods for comparison (e.g., how many singular values are removed when applying SVD and how they are selected, and similar for the rest of methods). Second, in the simulation results, although the metrics used show that the proposed method is better, the quality of the image is not improved compared to PCA and MS (as inferred from Fig. 6). Finally, in the real dataset results, the proposed method yields some kind of ringing effect in the middle of the dataset (see Fig. 8(g)). This behaviour is significantly weird and it is not commented at all by the authors. For all these reasons, I think the authors have not properly justified that the proposed method yields better results.

Furthermore, english writing must be definitely improved, as well as other typos (e.g., wrong referencing which results in Fig. ?? in line 162, wrong spelling of some words,...).

Round 2

Reviewer 1 Report

GPR B-scan clutter suppression based on subspace method is  a traditional but interesting question. This paper provides  a new decomposition and suppression algorithm, give some experiments to prove its effectiveness. In future, the combination of deep learning and signal processing may be more interesting at the aspect of low SNR B-scan images' denoising.
In Fig.1, the symbol "epsilon" is recommended to change to anther symbol, as "epsilon" represents the relative permittivity of soil in GPR field.

Author Response

Dear Prof,

Thank you for your comments concerning our manuscript. Those comments are valuable and very helpful. 
We have read through comments carefully and have made corrections. The symbol "epsilon" is replaced by “E ” mens energy. We also invited professional organizations to polish our manuscript grammar. Here, we attached revised manuscript in the formats of PDF for your approval.
What's more, deep learning is developing rapidly. Our next project plan will continue to focus on clutter suppression of through wall radar and ground penetrating radar, integrate traditional methods with deep learning, and explore more efficient methods. 

Sincerely,
Qibin Su.

Reviewer 2 Report

The Authors have provided an enhanced version of the manuscript, where further experiments were described. The Introduction was revised and several references were added to support the Authors' statements. The English language was also refined.

Overall, the quality of the manuscript has improved, although the novelty of the subject and the impact of the findings might be considered minor. I still suggest the Author a further revision of the English language, as I spotted some typos and uncommon definitions.

Author Response

Dear Prof,

  On behalf of my co-authors, we appreciate you very much for their positive and constructive comments and suggestions on our manuscript entitled “GPR Image Clutter Suppression Using Gaussian Curvature Decomposition in the PCA Domain”. (ID: remotesensing-1817556). 
  In this manuscript, a novel clutter suppression method principal component Gaussian curvature decomposition (PCGCD) is presented. First, the principal component analysis (PCA) is applied to divide the GPR B-scan data into different components. Then, a Gaussian curvature decomposition (GCD) method is proposed, which can be applied to subspaces in the PCA domain to retrieve more targets structure information from the random noise. Our PCGCD method obtained a higher PSNR and the highest SSIM, which is very advantageous for feature point detection and extraction of GPR images. 
  Thank you very much for giving us an opportunity to revise our manuscript. We appologize for the language problems in the original manuscript. We thoroughly checked and fixed grammatical errors in the final submission.The manuscript language has been thoroughly polished. We appreciate for your warm work earnestly, and hope that the correction will meet with approval. Once again, thank you very much for your comments and suggestions.

Sincerely,
Qibin Su.

Reviewer 3 Report

The authors have addressed my concerns and I think the article can be now published.

Author Response

Dear Prof,

  Thank you very much for your valuable comments during the review of round 1. Those comments are valuable and very helpful. We have read through comments carefully and have made corrections. Based on the instructions provided in your letter, we uploaded the file of the revised manuscript.

  We invited professional institutions to polish the language of the manuscript during this minor revisions. We would love to thank you for allowing us to publish the manuscript and we highly appreciate your time and consideration.

Sincerely,

Qibin Su.
